# Design of a MAPK signalling cascade balances energetic cost versus accuracy of information transmission

Alexander Anders[1,2,4], Bhaswar Ghosh[1,2,3,4 ✉], Timo Glatter[1] & Victor Sourjik [1,2 ✉]

Cellular processes are inherently noisy, and the selection for accurate responses in presence of noise has likely shaped signalling networks. Here, we investigate the trade-off between accuracy of information transmission and its energetic cost for a mitogen-activated protein kinase (MAPK) signalling cascade. Our analysis of the pheromone response pathway of budding yeast suggests that dose-dependent induction of the negative transcriptional feedbacks in this network maximizes the information per unit energetic cost, rather than the information transmission capacity itself. We further demonstrate that futile cycling of MAPK phosphorylation and dephosphorylation has a measurable effect on growth fitness, with energy dissipation within the signalling cascade thus likely being subject to evolutionary selection. Considering optimization of accuracy versus the energetic cost of information processing, a concept well established in physics and engineering, may thus offer a general framework to understand the regulatory design of cellular signalling systems.

[1] Max Planck Institute for Terrestrial Microbiology, 35043 Marburg, Germany. [2] LOEWE Center for Synthetic Microbiology (SYNMIKRO), 35043 Marburg, Germany. [3]Present address: International Institute of Information Technology, Gachibowli, Hyderabad, India. [4]These authors contributed equally: Alexander Anders, Bhaswar Ghosh. ✉email: bhaswar.ghosh@iiit.ac.in; victor.sourjik@synmikro.mpi-marburg.mpg.de

Living organisms have the ability to sense cues in the external and internal environment in order to initiate appropriate cellular responses. Inherent stochasticity of events involved in sensing and downstream transmission of signals may lead to noise, which is manifested as the variability in signalling outputs, e.g. morphological changes or reporter gene expression, across a population of clonal cells, leading to loss of information about the strength of the input[1–4]. While heterogeneity within a cell population can be beneficial in some cases[5], typical signalling pathways rather evolved to transmit information precisely in order to enable reliable input-dependent cellular responses.

The precision of signal decoding by cellular networks can be estimated using information theoretic approaches[6–11]. As the input (e.g. concentration of a chemical stimulant) varies and this change is transmitted via a signalling pathway, the change in the output (e.g. stimulated expression of a gene) carries information about the input variable. In general, the precision with which the input value can be estimated from measuring the output improves with larger changes (i.e. dynamic range) of the output and with lower output noise. According to Cramér–Rao inequality, the error in estimating the input from the output is bounded by the Fisher information[12], defined as the relative entropy change of the output distribution for an infinitesimal change in the input around a given input value.

Furthermore, information transmission capacity of signalling systems can also be estimated by calculating the mutual information[2,13], which is increasingly being used to characterize biochemical signalling networks[1,6,8–11,14]. The mutual information measures the mutual interdependence between the input and the output distributions by calculating the relative entropy of the output distributions conditioned on the input with respect to the unconditioned output distributions. In both cases, more information can be extracted about the input from the output distribution if the relative entropy change is large.

Cells apparently evolved several strategies to improve information transmission in the presence of the stochastic noise, using multiple genes[2] or multiple time points (i.e. time-averaging)[15] for readout or by actively suppressing noise[16,17]. The latter includes utilization of negative feedbacks, which have been shown both theoretically and experimentally to reduce signalling noise and therefore to enhance the information transfer[16–18]. Furthermore, negative feedbacks can reduce basal activity (i.e. activity in the absence of signal) and/or the activity of fully stimulated network—which can, dependent on the system, either increase or decrease the output range and thus information transfer[2,13]. Although increasing the accuracy of signalling typically imposes additional energetic cost[19–21], to which extent this fundamental trade-off between information and energy is reflected in the properties of cell signalling networks remains largely unclear.

We address this question by focusing on the negative feedback regulation within the pheromone response (or mating) pathway in *Saccharomyces cerevisiae*, one of the most studied examples of eukaryotic signalling. This pathway mediates communication and ultimately mating between two haploid mating types of *S. cerevisiae*, *MAT***a** and *MAT*α, which secrete a- and α-pheromones, respectively. Each pheromone is sensed by the opposite mating type, whereby perception of pheromone by a G protein-coupled receptor (Ste2 in *MAT***a**) stimulates the canonical mitogen-activated protein kinase (MAPK) cascade, leading to activation of the MAPKs Fus3 and Kss1[22–24]. Fus3, the major MAPK of the mating pathway, induces cell-cycle arrest, mating-specific changes in cell morphology and, via activation of the transcriptional activator Ste12, expression of mating genes including genes that encode components or regulators of the MAPK cascade[25–29]. Previous studies demonstrated that the pheromone pathway transmits information with high precision, as exemplified by a linear relationship between receptor occupancy and downstream responses ("dose–response alignment")[30] and by a uniform morphological transition of cell population into a mating-competent state ("shmooing") at a critical pheromone concentration[27,31]. Such uniformity in the output implies existence of, likely multiple, mechanisms to improve precision of information transmission within the pathway. Indeed, the negative feedback within the pheromone pathway provided by Sst2, a GTPase-activating protein (GAP) for the receptor-coupled G protein α-subunit, appears to fulfil such function[30,32].

Here, we primarily investigate the feedback regulation by Msg5, a dual-specific phosphatase for Fus3. Our experimental and computational analyses show that the pheromone-dependent transcriptional induction of Msg5 reduces variability and increases the dynamic range of the pathway output, hence increasing information transmission. Selection for increased accuracy could thus explain the presence of negative feedback regulation mediated by phosphatases as a common feature of the MAPK pathway topology. We further investigate the sensitivity of induction of the negative feedback provided by Msg5, which is activated at much higher pheromone dose than the upstream-acting negative feedback provided by Sst2[28,29]. We demonstrate that, while the precision of input estimation could be significantly enhanced by artificially increasing the sensitivity of *MSG5* gene induction, the naturally observed regulation appears optimal when considering energy investment into operation of the signalling pathway. We argue that such regulatory design, with lower induction sensitivity of the downstream feedback, might in general (optimally) balance accuracy of signalling against the energetic cost of pathway operation. Finally, we confirm experimentally that the phosphorylation/dephosphorylation cycle at the core of the MAPK signalling pathway has measurable fitness cost, thus likely placing it under evolutionary selection pressure.

## Results

**MSG5 and SST2 are induced at different pheromone doses**. Since we were interested in feedback regulation within the *S. cerevisiae* pheromone pathway that could be provided by transcriptional induction (i.e. transcriptional feedback regulation), we first determined which pathway-associated genes, and at what dose, are activated by the pheromone stimulation using transcriptomics (see "Methods"). Here, we exclusively focused on the branch of pheromone signalling mediated by MAPK Fus3, by utilizing strains devoid of the functional Kss1. Furthermore, all strains used for quantitative measurements of the α-pheromone responses were deleted for the α-pheromone-protease gene *BAR1* and for α-pheromone genes *MF*α*1* and *MF*α*2* to respectively avoid pheromone degradation or possible self-stimulation due to background expression of α-pheromone by *MAT***a** cells.

We find that, of the core pathway and regulatory components, expression of genes encoding the receptor Ste2, the GAP of the receptor-coupled Gα protein (Sst2), the kinase Fus3, its phosphatase Msg5 and the inhibitor of transcriptional activator Ste12 (Dig2) were upregulated (Supplementary Fig. 1), in a general agreement with the previous reports[28,29,33]. This suggests that the pheromone pathway is regulated by at least two pairs of positive/negative feedback loops, respectively, at the upper and lower levels of the cascade (Fig. 1a). We quantified induction sensitivities to pheromone of feedback regulators by determining EC$_{50}$ values, i.e. pheromone concentrations for achieving half-maximal induction (Fig. 1b). Notably, while both positive feedbacks (Ste2 and Fus3) and the upstream-acting negative feedback mediated by Sst2 become induced already at the low pheromone dose, the downstream-acting Msg5 negative feedback

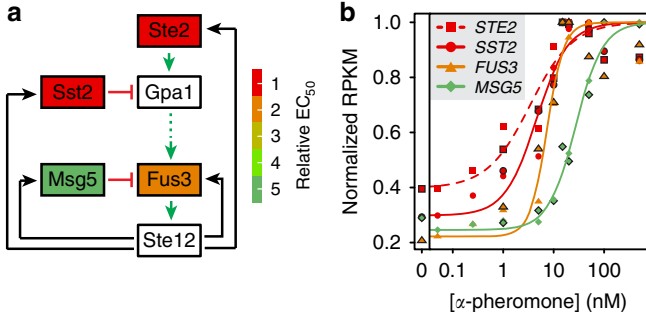

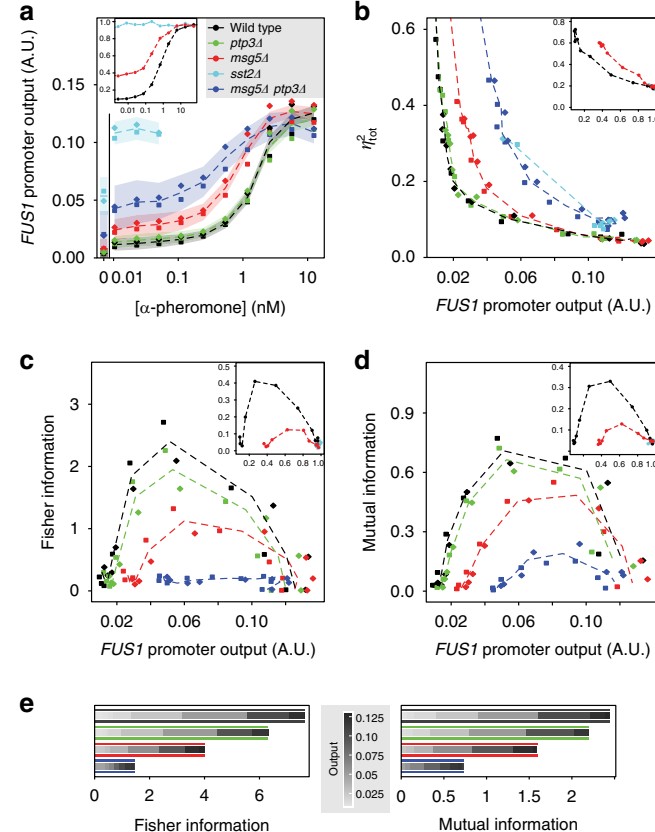

**Fig. 1 Activation of the pheromone response pathway induces multiple feedback loops. a** Simplified depiction of the pathway and its feedback regulators. Activation of the pheromone receptor Ste2 leads, through a signalling cascade, to activation of MAP kinase Fus3. Fus3-dependent phosphorylation of transcriptional activator Ste12 stimulates the expression of Ste2 and Fus3 as well as of two negative pathway regulators: Sst2, a GTPase activating protein (GAP) for Gα protein Gpa1; and Msg5, a phosphatase for Fus3. These upregulated components are coloured according to relative $EC_{50}$ values (normalized to the lowest value, approximately 4 nM for *STE2*) of their mRNA dose responses; mRNA levels of pathway components in white boxes were not upregulated. Black arrows denote transcriptional feedback regulation, green (dashed) arrows denote (indirect) activation and red blunt-end arrows indicate inhibiting activity. **b** Dose dependence of pheromone activation for the indicated genes encoding pathway components. Shown are normalized RNA levels as measured in two independent RNA sequencing experiments (symbols with and without frames, respectively) at 60 min after addition of respective dose of pheromone in a strain deleted for α-pheromone protease gene *BAR1* (Supplementary Table 1). Lines are fits with a sigmoidal function used to infer $EC_{50}$ values, used for the colour scale in **a**, **b**.

was upregulated only at much higher pheromone levels (Fig. 1b and protein colouring in Fig. 1a).

**Negative feedbacks improve information transmission**. We subsequently investigated the role of the negative feedbacks and their different induction sensitivities in the regulation of the pathway output. Since negative regulation by Sst2 has been already shown to suppress noise in the pheromone pathway[32], we focused on the negative feedback regulation by Msg5. To examine the noise-suppressing capability of Msg5, we measured the output fluorescence of the pheromone-responsive $P_{FUS1}$-GFP reporter in individual cells by fluorescence microscopy (see "Methods"). Indeed, we found that in the absence of either Sst2 or Msg5 the reporter output became more sensitive to low doses of pheromone (Fig. 2a). Moreover, the intercellular variability of the output in the population—total noise ($\eta_{tot}$), determined as the coefficient of variation (CV) of reporter activity—significantly increased (Fig. 2b). While the effect of *MSG5* deletion (*msg5Δ*) was mild compared to *sst2Δ*, it was further unmasked by deletion of phosphatase Ptp3. Ptp3 thus seems to partly compensate the loss of Msg5, although it is not transcriptionally induced by pheromone (Supplementary Fig. 1A) and in itself had nearly no impact on pathway response and noise (Fig. 2a). Another phosphatase, Ptp2, that was induced by the pheromone (Supplementary Fig. 1B) had virtually no effect on pheromone signalling (Supplementary Fig. 2). Besides increasing output noise, absence of either Msg5 or Sst2 lowered the threshold of pheromone stimulation and also elevated the basal pathway activity. Since the maximal output responses remained nearly unaffected, these deletions thus effectively reduced the output range.

In order to quantify signalling accuracy in wild type and mutant strains, we invoked an information theory approach. As

**Fig. 2 Negative feedback regulators improve accuracy of input estimation and information transmission. a** Dose dependence of pathway activation, measured using activity of a $P_{FUS1}$-GFP pathway reporter, for wild type (black) and strains deleted for negative regulators (colours as indicated). Data were collected between 140 and 210 min after pheromone addition, using time-lapse microscopy (see "Methods"), in two independent experiments (shown with different symbols). Dashed lines connect means for both experiments and serve as a guide to the eye; shaded areas centred on those lines show cell-to-cell variabilities (s.d.) across the cell populations, again averaged over both experiments with at least 300 cells per point and experiment. **b** Pathway noise, calculated as the coefficient of variation of $P_{FUS1}$-GFP levels across the population, in the individual experiments (symbols as in **a**), plotted against the $P_{FUS1}$-GFP output. Dashed lines connect the means of noise and $P_{FUS1}$-GFP output for both independent experiments. **c**, **d** Fisher (**c**) and mutual (**d**) information, calculated as described in "Methods" independently for both experiments (symbols are as in **a**). Notably, although mutual information was calculated between the pheromone input and the pathway output, it was plotted against the pathway output at a particular pheromone input for better comparison between strains that have different sensitivities to pheromone. Dashed lines connect means for both independent experiments. Fisher and mutual information are not shown for *sst2Δ* strain. Insets in **a**–**d** show results of stochastic simulations using a simplified mathematical model of the pheromone pathway (see main text). **e** Aggregated, i.e. summed up over the whole output range, Fisher (left) and mutual (right) information, with individual contributions at different output levels shown in different shades of grey. Aggregated information was calculated from means for both independent experiments. Edges of bars are coloured according to the strains as indicated in **a**.

mentioned above (see "Introduction"), both output range and pathway noise impact the signalling precision and amount of information that can be transmitted through a signalling pathway[2,13,16,30]. Precision in the input estimation can be characterized locally (i.e. at different pathway outputs) by the

Fisher information[12,34] (see Supplementary Note 1 for more details). Fisher information was indeed significantly higher, virtually over the whole range of pathway outputs, for the wild type as compared to *msg5Δ* or *msg5Δ ptp3Δ* strains (Fig. 2c). Consequently, aggregated Fisher information was highest for the wild type, too (Fig. 2e, left). In order to distinguish respective contributions of increased output range and decreased noise to the Msg5-mediated improvement of information transmission, Fisher information was also calculated when projecting noise levels of the wild type onto *msg5Δ* dose responses and vice versa. We found that individual contributions of the output range expansion and noise suppression by Msg5 to the improved information transmission were roughly equal and synergistic (Supplementary Fig. 3). Thus negative feedbacks improve accuracy of input estimation by both expanding the dynamic range of the output and lowering the output noise.

To confirm these results obtained using Fisher information, we also calculated mutual information, another frequently used measure of information transmission[35]. To directly compare these two measures of information transmission, mutual information was calculated locally at a particular level of pathway stimulation, as done previously[14] (Supplementary Note 1). Relative differences in transmitted mutual information between strains were comparable to those observed for Fisher information (Fig. 2d, e, right panel). Notably, similar differences between strains were observed at other time points of stimulation for both Fisher and mutual information (Supplementary Fig. 4).

Finally, we tested whether it is the negative transcriptional feedback or simply the negative regulation of the pathway that are important for enhanced precision, by constitutively expressing *MSG5* under control of a doxycycline-inducible promoter. Such constitutive Msg5 production could only partially rescue the output range, noise suppression and precision (Supplementary Fig. 5), implying that pathway-dependent induction of the transcriptional feedback is indeed crucial.

**Sensitized *MSG5* induction improves information transmission.** Although both Sst2 and Msg5 negative feedback regulators improve information transmission, they display different induction sensitivities, with Sst2 that acts upstream in the pathway being induced at lower dose than the downstream-acting Msg5. In order to study the impact of the dose dependence of feedback

induction in silico, we constructed a simplified mathematical model of pheromone signalling that consists of a two-step phosphorylation cascade activated by pheromone at the upper level and incorporates two negative (Sst2 and Msg5) as well as two positive (Fus3 and Ste2) feedbacks (Supplementary Fig. 6A, see "Methods" and Supplementary Note 2 for details). Simulations of this model qualitatively confirmed that negative feedbacks reduce basal activity and noise of the pathway output, therefore enhancing Fisher and mutual information (Fig. 2a–d insets).

We then used this model to investigate the dependence of the signalling accuracy on induction sensitivity of the Msg5 feedback by systematically varying $K_D$ value that quantifies binding strength of active Ste12 (i.e. phosphorylated Ste12-P) to *MSG5* promoter and calculating resulting dose responses to pheromone, output noise and Fisher information (Fig. 3a, Supplementary Fig. 6B, C). Interestingly, simulations predicted that maximum information would be transmitted by the pathway at the highest sensitivity of *MSG5* induction (Fig. 3a), in contrast to the experimentally observed low induction sensitivity.

To test this model prediction experimentally, we replaced the native promoter of *MSG5* with $P_{FUS1}$ promoter that is induced by lower levels of pheromone (Supplementary Fig. 1). The modified induction of *MSG5* both decreased basal pathway activity and improved suppression of noise, particularly in the lower range of activity (Fig. 3b, c), similar to the results of model simulations (Supplementary Fig. 6B). Consequently, the overall information transmission increased, while its range slightly shifted to lower activity as compared to the wild type (Fig. 3d), again confirming model predictions. We obtained similar results for high-sensitivity induction of *MSG5* by other pathway-dependent promoters, in these cases measured using flow cytometry (Supplementary Fig. 7). Notably, our analysis showed that both lowered basal pathway activity and reduced noise at low pathway activity contributed to the observed increase in Fisher information (Supplementary Fig. 8).

**Sensitive feedback induction maximizes information.** We next systematically investigated the dependence of information transmission through the pathway on the induction sensitivities of both negative feedbacks, Sst2 and Msg5. Since in our simulations information transmission to the downstream transcriptional

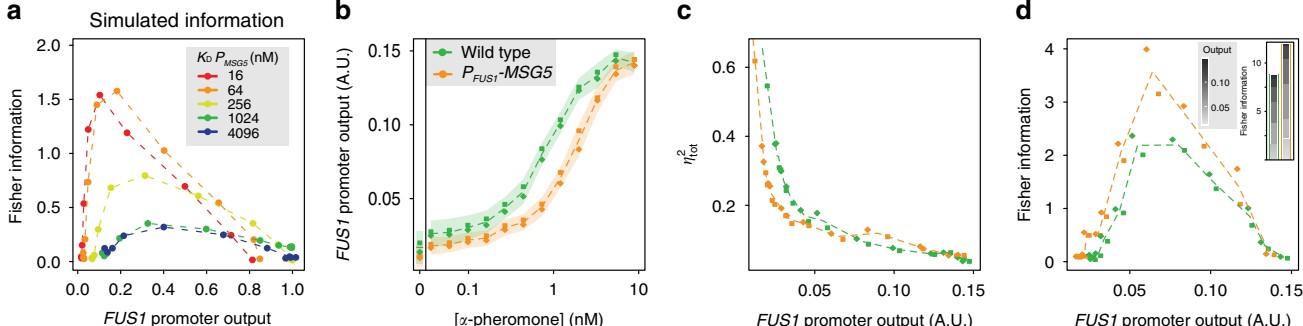

**Fig. 3 Sensitized MSG5 induction leads to improved input estimation. a** Dependence of simulated Fisher information on sensitivity of *MSG5* feedback induction, altered by adjusting the parameter defining binding affinity of active Ste12-P to the *MSG5* promoter ($K_D P_{MSG5}$). For simulation of the wild type in Fig. 2, $K_D$ was 700 nM. **b–d** Effect of sensitized *MSG5* feedback induction on dose dependence of the pathway response (**b**), noise (**c**) and Fisher information (**d**). *MSG5* induction was sensitized by replacing its native promoter with $P_{FUS1}$ promoter that responds with higher sensitivity to pheromone. Data were acquired between 110 and 170 min after pheromone addition in two independent experiments (shown with different symbols in **b–d**). Dashed lines serve as guides to the eye and connect means for both experiments; shaded areas in **b** centred on those lines show cell-to-cell variabilities (s.d.) across the cell populations, again averaged over both experiments. Similar results were observed when *MSG5* induction was sensitized using other promoters (Supplementary Fig. 6). Inset in **d** shows aggregated Fisher information calculated with means for both experiments.

reporter is reduced by saturation of the phosphorylation and DNA-binding functions of Ste12 (Supplementary Fig. 9A), we calculated total information (i.e. information averaged over all simulated inputs) based on active Fus3 (i.e. double-phosphorylated Fus3-PP) as the output. Also at this level, the information transmission was reduced in the absence of negative feedback regulators Sst2 and Msg5 (Supplementary Fig. 9B). Consistent with simulation of the transcriptional output (Fig. 3a), maximal aggregated (total) transmission information contained in Fus3-PP was observed at similarly high induction sensitivities (i.e. low $K_D$ values) for Msg5 and Sst2 (Fig. 4a, Supplementary Fig. 10). Notably, similar to our experiments, varying $K_D$ values in the simulations not only changed the sensitivity of induction (i.e. $EC_{50}$ value) but, at the same time, also altered dynamic ranges of the analysed output.

**Information transmission is balanced by energy consumption.** Since both experiments and modelling suggest that induction sensitivity of the Msg5-mediated negative feedback in the

pheromone signalling pathway may not to be optimized for maximizing information transmission, we asked in which respect the naturally observed design might be superior to a potential information-maximizing design. While having the capacity to reduce noise and/or increase the output range, negative (feedback) regulation requires energy. In the cases of Msg5 and Sst2 feedbacks, this means consumption of adenosine triphosphate (ATP) in cycles of phosphorylation and dephosphorylation (subsequently simply referred to as phosphorylation cycles) and increased consumption of guanosine triphosphate (GTP) for activation and de-activation cycles of receptor-coupled G protein, respectively. Generally, maintaining a given pathway output in the presence of negative (feedback) regulation entails higher energy consumption than its absence, which might constrain optimization of negative feedback regulation. We estimated that the rate of ATP consumption that is required to maintain half of total Fus3 molecules phosphorylated[36], ~$10^4$ molecules/cell at full pathway induction, amounts to 0.2% of total cellular energy use in rich medium (Supplementary Note 5). Thus the cost of energy investment in operating the MAPK signalling pathway might be

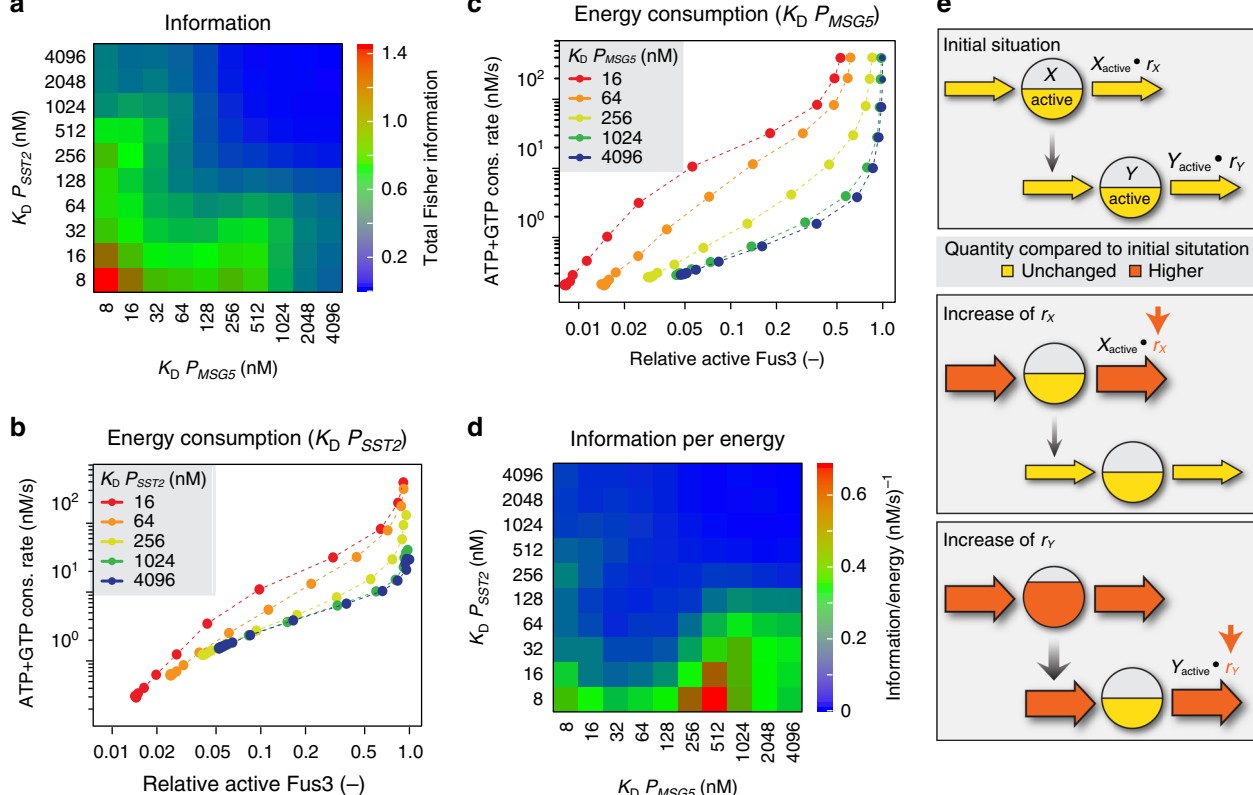

**Fig. 4 Induction of negative feedback regulators may balance accuracy versus energy investment. a** Simulated dependence of pathway accuracy (information) on sensitivities of *SST2* and *MSG5* induction. The heat map shows total Fisher information of active Fus3 (Fus3-PP) over the same $5 \times 10^6$ range of stimulus strength (see main text and Supplementary Fig. 8 for details) for varying binding affinities ($K_D$) of Ste12-P to *SST2* and *MSG5* promoters. **b, c** Simulated energy (ATP+GTP) consumption rates as a function of relative active Fus3 (Fus3-PP normalized to maximum per plot) for different induction sensitivities of *SST2* (**b**) and *MSG5* (**c**). Binding affinities of Ste12-P to *SST2* and *MSG5* promoters were changed individually while keeping affinity to the respective other promoter fixed at 8 nM. **d** Simulated dependence of information per energy on sensitivities of feedback induction. The heat map shows total Fisher information per energy (see text for details) simulated as in **a**. **e** Cartoon illustrating difference between energetic costs of negative feedback regulation at two stages of a simulated signalling cascade. Here, activities of two consecutive positive regulators (*X* and *Y*), indicated by filling heights of the respective circles, are determined by the influx (e.g. phosphorylation) and outflux (e.g. dephosphorylation) rates of energy. Influx rates at the upper (lower) level depend on the signal strength (e.g. pheromone stimulation) and activity of the upstream regulator, respectively. Outflux rate constants ($r_X$, $r_Y$) correlate with the activities of negative regulators (e.g. phosphatases). Red colour indicates increase in quantities compared to the initial situation (upper panel). In order to sustain constant cascade output $Y_{active}$ while increasing $r_x$ (middle panel), the increased outflux requires compensation by higher influx into *X* (i.e. stronger stimulation is required to achieve the same output). In this case, increased flux of energy solely at the affected level is sufficient for compensation. However, when increasing $r_Y$ (lower panel), the compensatory higher influx into *Y* additionally requires an increase in $X_{active}$, which in turn requires higher influx into *X*. Thus compensation of increased $r_Y$ entails increased energy fluxes on both cascade levels.

sufficiently high to affect cellular fitness. This estimated cost of continuous phosphorylation cycle of Fus3 is comparable to the cost of cellular investment into the biosynthesis of signalling proteins (Supplementary Note 5), reminiscent of signalling in bacterial chemotaxis[19], and thus, indeed, might be subject to evolutionary optimization.

Consequently, in our model simulations, we took into consideration the energy consumed during the signalling process, GTP at the upper level and ATP at the lower level. Intuitively, in order to maintain a given pathway output, higher activity of a negative regulator requires stronger pathway activation by the pheromone, which increases turnover of GTP and/or ATP. Induction of a negative feedback regulator already at low levels of pathway stimulation should result in increased energy consumption over a wide range of pathway activities. This effect is indeed apparent when varying induction sensitivities of both *SST2* (Fig. 4b) and *MSG5* (Fig. 4c). However, while the increase in energy consumption with increased sensitivity is relatively modest for *SST2*, it is much more pronounced for *MSG5*. To take into account both information transmission and energy consumption in our simulations, we calculated integrated information per energy[37], or energy efficiency, for different sensitivities of feedback induction:

$$\text{Efficiency} = \int_{\theta_1}^{\theta_2} \frac{F(\theta)}{E(\theta)} P(\theta) \mathrm{d}\theta. \quad (1)$$

Here, $F(\theta)$ is the Fisher information, $E(\theta)$ is the energy expenditure and $P(\theta)$ is the probability of the input (Supplementary Notes 1 and 4). This definition is convenient in restricting our energy calculation to the range of pathway stimulation that contains information. In contrast to the maximal information transmission (Fig. 4a), maximal information per energy was obtained at high-sensitivity *SST2* and low-sensitivity *MSG5* induction (Fig. 4d and Supplementary Note 4), which is consistent with our experimental observations. This finding indicates that the natural design of the MAPK pathway might be tuned to balance the precision of information transmission with energy investment needed to enable high precision. More generally, our analysis suggests that for signalling cascades that include negative feedbacks acting at different levels of the cascade—a common feature of eukaryotic signalling[38,39]—induction of downstream-acting regulators with lower sensitivity than those acting upstream might be fundamentally beneficial from an energetic perspective (Fig. 4e, see "Discussion").

**Cycling of MAPK phosphorylation reduces growth fitness.** While theoretically deducible, the effect of energy consumption within the signalling cascade on cellular fitness has not been shown experimentally. Given its small cost in comparison to total cellular energy consumption, direct measurement of the phosphorylation-cycle effect on cellular energy turnover would not be feasible. However, the effect could become measurable as growth disadvantage under long-term competition with continuous activity of the phosphorylation cycle. We hence performed such experiment by using two different enzymatically inactive Fus3 variants, one of which could, however, still be phosphorylated (Fus3-K42R) while the other could not (Fus3-KTY) (Fig. 5a and Supplementary Fig. 11A)[40]. Since the background strain used for our experiments lacks functional Kss1 (see above), replacement of wild-type Fus3 with these inactive variants completely abolished transmission of the pheromone signal further downstream (Supplementary Fig. 11B, C). Thus comparing growth of strains expressing either of the two Fus3 variants along with mNeongreen or mTurquoise to allow their discrimination

enabled us to selectively assess the effect of pheromone-dependent Fus3 phosphorylation cycle.

To confirm that Fus3-K42R becomes phosphorylated in a pheromone-dependent manner in vivo[40], we identified the corresponding phosphorylated peptide using mass spectrometry (Supplementary Fig. 12). Indeed, the degree of phosphorylation of this peptide was higher in the presence of pheromone (Supplementary Fig. 12). Moreover, the phosphorylation was also elevated in the absence of Msg5, suggesting that K42R is dephosphorylated by Msg5.

When the MAPK pathway activity was stimulated by pheromone, the relative abundance of strains carrying Fus3-K42R steadily decreased in the co-culture with Fus3-KTY-expressing cells (Fig. 5b, see Supplementary Fig. 13 for individual experiments). The rates of growth divergence were, depending on growth conditions, between about 0.02 and 0.07 per day (insets in Fig. 5b–d). Assuming a constant doubling time of 100 min, and thus a growth rate of approximately 10 day$^{-1}$, this translates into about 0.2–0.7% growth fitness defect of Fus3-K42R- compared to -KTY-expressing cells in the presence of pheromone. This decrease did not depend on which of the strains was labelled with mNeongreen or mTurquoise, confirming that labelling itself introduced no bias in growth (also apparent in the co-culture of two differently labelled *fus3Δ* strains, Fig. 5a). Furthermore, we performed control experiments to assess whether experimental results were in line with our interpretation that indeed phosphorylation cycle of Fus3-K42R was causing the observed pheromone-dependent growth deficit. Under this assumption, we expected (i) absence of phosphorylation cycle with deletion of Fus3-phosphorylating kinase Ste7 and thus equal growth of strains expressing Fus3-K42R and -KTY in the presence of pheromone and (ii) slow-down of phosphorylation cycle in the absence of phosphatase Msg5 due to decreased dephosphorylation rates and thus reduction of growth difference between Fus3-K42R and -KTY cells compared to competition between Msg5-expressing cells. We tested both predictions and indeed could confirm them experimentally (Fig. 5c, d). Taken together, we hypothesize that pheromone-dependent growth deficiency of cells expressing Fus3-K42R is caused by the energetic burden of its phosphorylation cycle and, consequently, that energetic cost of enhanced information transmission was subject to evolutionary selection, shaping the negative feedback design of the yeast MAPK pathway.

## Discussion
Cells need to reliably respond to the external cues in spite of the inherent stochasticity of the events involved in sensing and downstream signalling. This stochastic noise may lead to the intercellular variability of responses, and single-cell measurements indeed revealed that expression of the same gene can be highly variable across apparently homogeneous cell populations[3,4]. Besides intrinsic stochasticity of individual processes, the observed variability in gene expression might also have extrinsic sources, e.g. variable levels of global transcription and translation factors[41].

Negative feedbacks have been shown to reduce cellular noise and thus enhance information transmission capacity of signalling pathways[16–18]. This includes noise suppression through negative feedback regulation provided by Sst2 within the yeast pheromone signalling cascade[30,32]. Furthermore, negative feedbacks can suppress pathway activity both in the absence and/or presence of stimulation[2]. Dependent on the specific parameters of a signalling pathway, suppression of the basal activity can lead to enhanced response output range and thus information transmission[2,13].

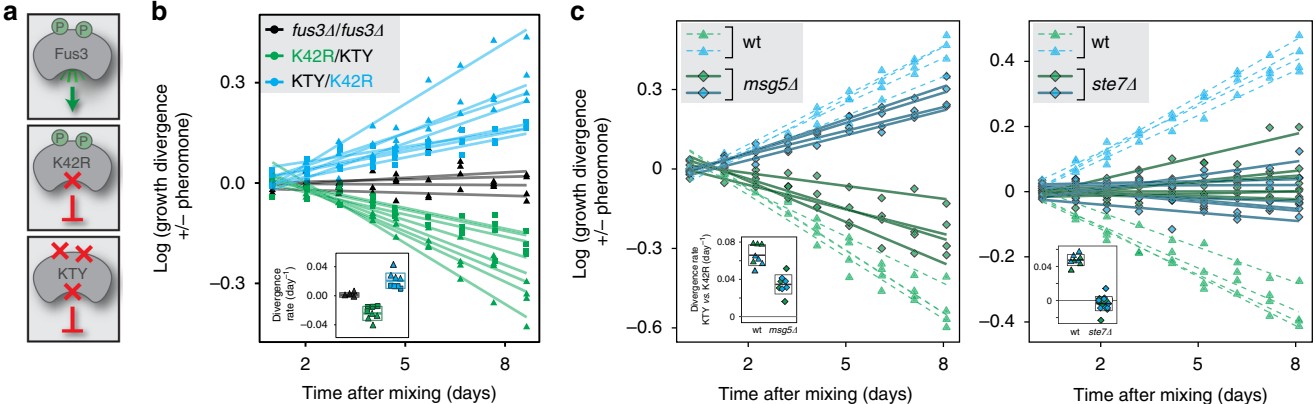

**Fig. 5 Continuous dephosphorylation/phosphorylation of enzymatically inactive Fus3 lowers competitive growth fitness. a** Schematic depiction of employed Fus3 variants. A single amino acid replacement renders Fus3-K42R and -KTY enzymatically inactive and thus incapable of transmitting the signal. Fus3-KTY additionally carries amino acid replacements at both phosphorylation sites. **b** Fitness cost of phosphorylation cycle, monitored over time as the ratio between mNeongreen (Ng)- and mTurquoise (Tq)-expressing cells grown in a co-culture in the presence of pheromone (20 nM), normalized to the corresponding ratio in absence of pheromone. Different colours depict co-cultures of competing strains that carry different *fus3* alleles and fluorescent markers: *fus3Δ*+Ng vs. *fus3Δ*+Tq (black), *fus3-K42R*+Ng vs. *fus3-KTY*+Tq (green) and *fus3-KTY*+Ng vs. *fus3-K42R*+Tq (blue). For each pair, four co-cultures per genotype with independent transformants were tested. Co-cultures were grown for several days with re-inoculation twice a day and flow cytometric measurements once a day to determine ratios of Ng- to Tq-expressing cells. Combinations of *fus3-K42R*- and -KTY-expressing cells were analysed in two independent experiments (depicted by different symbols), while control of *fus3Δ* vs. *fus3Δ* was analysed in one experiment. Lines are linear fits for individual co-cultures. Slopes derived from these fits quantify the rate of divergence of Ng/Tq ratios between co-cultures grown in the presence and absence of pheromone (*Inset*). Centres and boundaries of boxes in the *Inset* depict means +/− s.d.; colouring for different co-cultures is according to the main plot. In pairwise comparisons using two-sided *t* test, *p* value was 7.08e−5 with 95% confidence interval (CI) ranging from 0.0169 to 0.0338 for *fus3Δ*-*fus3Δ* (black, n = 4 biologically independent samples) against *K42R*-*KTY* co-cultures (green, n = 4 biologically independent samples examined over 2 independent experiments) and *p* value was 9.0e−4 with CI from −0.0287 to −0.0105 for *fus3Δ*-*fus3Δ* (black) against *KTY*-*K42R* co-cultures (blue, n = 4 biologically independent samples examined over 2 independent experiments). **c** Fitness of mutants with reduced dephosphorylation rate (*msg5Δ*, left) or abolished phosphorylation (*ste7Δ*, right) compared to wild type. Wild-type pairs (wt, symbols and colours as in **b**, dashed lines for linear fits) in co-cultures were same as in **b**. Each of those strains was subjected to gene deletion and the resulting deletion strains (one and two clones per parental strain for *msg5Δ* and *ste7Δ*, respectively) were grown in co-cultures (diamond symbols, solid lines for linear fits) in a competition experiment alongside parental/wild-type co-cultures. Insets show divergences of ratios between *fus3-KTY*- and -*K42R*-expressing cells between co-cultures grown in the presence or absence of pheromone, as derived from the slopes of linear fits shown in the main figures. Centres and boundaries of boxes in the inset depict means +/− s.d. In pairwise comparisons with two-sided *t* test, *p* value was 4.62e−5 with CI from 0.0198 to 0.0429 for wild type (n = 8 biologically independent samples) against *msg5Δ* (n = 8 biologically independent samples) co-cultures and *p* value was 4.32e−12 with CI from 0.0445 to 0.0577 for wild type against *ste7Δ* (n = 16 biologically independent samples) co-cultures. To increase throughput, co-cultures were grown here in 96-well plates without shaking, which might explain slightly higher absolute divergence rates in these experiments compared to **b** where co-cultures were grown in 24-well plates with shaking.

In general, negative feedback regulation entails dissipation of energy and energy requirements for accurate information transmission in signalling cascades have been emphasized in several studies[19–21,42–44]. Nevertheless, the trade-off between potentially conflicting optimization of negative feedbacks for energy dissipation and for information transmission remained unexplored. In this study, we demonstrate that negative feedback mediated by the phosphatase Msg5 reduces the pathway noise and amplifies the response output range, thus improving the precision of input estimation and the information transmission capacity of the signalling cascade, as measured by Fisher and mutual information, respectively.

We observed that, whereas Msg5 at a lower level of the cascade complements the Sst2 negative feedback at the upper level, *MSG5* transcription is induced at much higher dose of the input signal than that of *SST2*. This seeming correlation between the order of negative feedback induction and the cascade stage could not be simply rationalized in the context of information transmission, since both our simulations and experiments showed that similar and sensitive induction of both *SST2* and *MSG5* improves the information capacity of the cascade. Furthermore, according to our simulations, information was largely symmetric regarding the induction order of both feedback regulators, meaning that the precision of input estimation was virtually the same regardless of

*SST2* being induced with higher sensitivity than *MSG5* or vice versa. However, a striking difference between high- and low-sensitivity induction of *MSG5* was observed when energy consumption during signalling was additionally considered in our mathematical model, with the latter configuration consuming less energy for achieving the same output. Thus both high-sensitivity induction of both negative feedbacks as well as a reversed order of induction are energetically inferior to the observed induction order. More generally, our analysis indicates that, for signalling cascades that include negative feedbacks acting at different levels of the cascade, induction of downstream regulators might be generally energetically more expensive than those acting more upstream (Fig. 4e). Notably, we further demonstrate experimentally that futile cycling of Fus3 phosphorylation and dephosphorylation indeed can bear measurable fitness cost. Thus dissipation of energy at this step likely resulted in evolutionary selection for trade-off optimization between information and energy consumption.

Notably, instead of using the more conventional method of calculating channel capacity over all possible inputs[2], here we used metrics that enabled us to quantify information at every input, which was necessary for defining the cost of energy consumption for information transmission at a particular input. We hence primarily used Fisher information, since it is naturally

defined for a particular input. Nevertheless, we also calculated local mutual information as done previously[14], thus allowing us to confirm that both information metrics give qualitatively similar results.

Here, our analyses focused on Msg5 and, to a lesser extent, on Sst2, the two major negative transcriptional feedback regulators in the pheromone signalling pathway. Although Dig1, an inhibitor of Ste12, has been previously reported to suppress noise in basal pathway activity[45], we observed no pheromone-dependent transcriptional induction of *DIG1* and thus did not consider it as a bona fide negative transcriptional feedback. And, while closely related regulator Dig2 was indeed transcriptionally induced at high levels of pheromone stimulation, its exact role in the pathway regulation remains unclear.

Taken together, our analysis provides evidence that evolutionary selection on signalling networks not only maximizes precision of information transmission but also minimizes the corresponding energy investment. Such trade-offs between multiple (conflicting) objectives may be the fundamental principle in many biological system designs[46]. Although individual details of the cost-accuracy trade-off will likely depend on the importance of signalling accuracy for fitness in a specific system, this trade-off is likely to play a major role in the evolution of signalling and thus to shape the architecture of cellular signalling networks.

## Methods

**Strains and growth conditions**. *S. cerevisiae* strains used in this study are of mating type *MAT***a** and derivatives of SEY6210a (*MAT***a** *leu2-3,112 ura3-52 his3Δ200 trp1Δ901 lys2-801 suc2Δ9*). Notably, this strain caries loss-of-function mutations in the *kss1* gene (our unpublished data), allowing us to exclusively analyse the major, Fus3-mediated signalling branch of the pheromone pathway. Generally, strains used for quantitatively measuring responses to α-pheromone were deleted for α-pheromone-protease gene *BAR1* and α-pheromone genes *MFα1* and *MFα2*. All strains are listed in Supplementary Table 1.

Generally, the synthetic defined media (SD or LoFlo-SD) for growing yeast in liquid were composed of yeast nitrogen base (YNB, Formedium) or low-fluorescence YNB (LoFlo-YNB, Formedium) with complete supplement mix (Formedium) and 2% glucose. Routinely, cells in liquid media were incubated overnight in an orbital shaker at 200 rpm, diluted 1:50 to 1:100 into fresh media the following morning and grown to mid-exponential phase prior to starting the experiment.

For RNA sequencing experiments, strain yAA95 was grown in 500 ml SD at 30 °C. At mid-exponential phase (OD$_{600}$ approximately 0.5), pheromone stimulation was initiated by transferring aliquots of the suspension into separate flasks with prepared stock solutions of synthetic α-pheromone (Sigma). After incubation for 60 min, cells were harvested by transferring 5 ml of suspension into 15-ml tubes containing 5 ml ice and mixed to immediately cool down the suspension. Further steps were carried out on ice or at 4 °C. After centrifugation and removal of supernatant, cell pellets were re-suspended in ice-cold water and transferred to 2-ml Eppendorf tubes. After another centrifugation step, the supernatant was carefully removed, and the cell pellets were stored at −20 °C until further processing. For the time-course experiments, the harvesting procedure was repeated at different time points.

For microscopic experiments, cells were grown in LoFlo-SD with 2 μM casein at 25 °C. Where applied, doxycycline was already present during overnight growth and was kept at the same concentrations for day cultures and during microscopy. Note that addition of casein results in higher apparent sensitivities of cell responses to pheromone likely due to prevention of pheromone adsorption to surfaces[47]. Thus absolute sensitivities reported here are higher for microscopy compared to RNA sequencing experiments.

For growth-competition experiments, growth media were composed of LoFlo-SD (experiment 2 in Supplementary Fig. 13) or SD (all other experiments) with 2 μM casein and supplemented with 20 μg/ml doxycycline and/or 20 nM α-pheromone where applicable. Cells were grown in 24-well plates (Greiner Bio-One) in 1 ml medium at 30 °C with shaking at 200 rpm and diluted twice a day (in the morning and evening) 1:50 into fresh medium of the same composition. For higher-throughput experiments with wild type and deletion strains (Fig. 5c), cells were grown in 96-well plates (Greiner Bio-One) in 250 μl medium at 30 °C without shaking. Again, cells were diluted twice a day 1:50 into fresh medium.

**Fluorescence time-lapse microscopy and analysis**. The microscopic assay[48] is illustrated in Supplementary Fig. 14. Briefly, images were acquired on a wide-field fluorescence microscope (Nikon Ti-E) equipped with a solid-state white-light light-emitting diode source (Sola SE-II), a motorized stage, a ×40 dry objective (Nikon

Plan Apo ×40 Lambda, 0.95 N/A), a sCMOS camera (Andor Zyla) and an incubator with heater controller (Digital Pixel). The green fluorescent protein (GFP) signal was acquired using a 470/40 nm excitation filter and a 525/50 nm emission filter, and the mCherry signal was acquired with 575/25 nm and 647/57 nm, respectively. Cell suspensions were transferred to a 96-well glass-bottom plate (Greiner Bio-One) coated with type-IV Concanavalin A (Sigma-Aldrich), and addition of α-pheromone stock solutions was performed after allowing cells to settle down for approximately 30 min. Image acquisition was started immediately after addition of pheromone and repeated periodically at defined time intervals over the course of several hours at 25 °C.

Single-cell median values of fluorescence intensities were extracted from the images with the freely available software CellProfiler version 2[49] by performing cell segmentation on the bright-field images and using the defined masks for measuring intensities in the corresponding fluorescence images. Raw data extracted by CellProfiler were further processed using the statistical software R version 3[50]. Upper and lower three percentiles per field of view and time point in either fluorescence channel were routinely removed from further analysis. Fluorescence intensities of GFP-only and mCherry-only expressing strains were used to correct fluorescence intensities of single cells for autofluorescence and bleed-through between the fluorescence channels. The corrected values were used for further analysis. The final number of analysed cells per well and time point was between 300 and 4000. Original images and tables containing single cell data are available on request.

**Flow cytometric measurements**. Cell suspensions were injected from a 96-well plate (Greiner Bio-One) with a high-throughput sampler into an LSR Fortessa Special Order flow cytometer (BD Biosciences). Fluorescence was measured with lasers of different wavelengths: 488 nm for GFP, 447 nm for mTurquoise (Tq), 561 nm for mCherry, and 514 nm for mNeongreen (Ng). Using the BD FACS DIVA software (BD Biosciences), cells were gated in a FSC-A/SSC-A plot to exclude debris. Ten thousand cells from within the gate were acquired per sample. For growth-competition experiments, cells were gated in a Ng/Tq plot to determine the relative amounts of Ng- and Tq-expressing cells in the co-culture.

**RNA preparation and sequencing**. Frozen cell pellets were re-suspended in 500 μl ProtK buffer (100 mM Tris/Cl pH 7.9, 150 mM NaCl, 25 mM EDTA, 1% sodium dodecyl sulfate) with freshly added 100 μg/ml Proteinase K (ThermoFisher). Five hundred microlitres of glass beads were added, and cells were disrupted by strong vortexing for 5 min. The resulting lysate–glass bead mixture was incubated for 60 min at 37 °C. The RNA was isolated by 25:24:1 aqua-phenol:chloroform:isoamyl alcohol (Carl Roth) extraction followed by a chloroform extraction and precipitated with ethanol. The RNA pellet was washed once with 75% ethanol, re-suspended in water and treated with RNase-free DNaseI (Roche Life Science) according to the manufacturer's protocol. After another precipitation/wash step, the RNA was dissolved in nuclease-free water and analysed for integrity on a formaldehyde/agarose gel. The RNA was depleted of ribosomal RNAs with the Ribo-Zero Gold rRNA Removal Kit (Illumina) and reverse-transcribed with random hexamers at the deep-sequencing facility at BioQuant, Heidelberg. Sequencing was performed at the GeneCore facility at EMBL Heidelberg, with 50 bp read length, single-end reads and 9 samples per lane, by Illumina (Solexa) sequencing.

**Analysis of RNA sequencing data**. Sequence reads were mapped onto the yeast genome (ensemble *S. cerevisiae* genome) with Bowtie 2[51]. The resulting strand-specific read densities were used to calculate the RPKM (reads per kilobase of transcript, per million mapped reads) values for all annotated genes (ensemble *S. cerevisiae* genome). A threshold of twofold higher (lower) expression than in the non-pheromone-treated sample was used for designating genes as being induced (repressed) genes. The dose–response curves were generated from the RPKM values at different pheromone concentrations, and EC$_{50}$ values were determined by fitting the dose–response curves using a sigmoidal function.

**Calculation of noise and information**. Time-lapse single-cell measurements of the $P_{FUS1}$-GFP reporter were used to calculate the CV, used as a measure of the total noise of gene expression. The accuracy of signal transduction was determined using Fisher information, which was calculated by smoothing the dose dependence of the response and of standard deviation with a spline interpolation in order to obtain the derivatives of the log-likelihood function at different inputs. The distribution of the outputs at each input was assumed to be normal to numerically calculate the integral for taking average of the derivatives over the distribution function (Supplementary Note 1). Information transmission was further quantified by calculating mutual information at every input by taking a sliding window of three inputs. Mutual information within this window was estimated by binning the distribution of the output values in 20 bins to calculate the conditional and unconditional entropy (Supplementary Note 1). Numerical integration for the Fisher information and the entropy calculation for the mutual information were carried out using the Package entropy and the spline function in R software.

**Mathematical model of the pheromone response pathway**. We considered a two-step phosphorylation cascade model of the pathway (Supplementary Fig. 6A),

where the receptor Ste2 activates kinase Fus3 and the activated Fus3 further induces the activity of the transcription factor Ste12, which in turn stimulates the transcription of the *FUS1* gene. The system is modelled as a set of ordinary differential equations (ODEs; Supplementary Note 2)[32,39,52,53], and noise is introduced by randomly selecting production rates as well as initial concentrations of the pathway components from a log-normal distribution of values. The simulations were performed by solving the ODEs using CVode[54] interfaced with MATLAB in a population of 500 cells for 1000 s. By default, simulations covered a pheromone-concentration range of $3^{14}$-fold with serial 3-fold increases, usually starting with $10^{-4}$ nM and leading up to approximately 480 nM. Noise, Fisher information and mutual information were calculated from the simulated data following the same methods as in the case of the experimental data (see above). We considered that energy is consumed in signal transduction at two levels: at the receptor level in the process of GTP hydrolysis during inactivation of the receptor by Sst2 as well as in the two-stage phosphorylation reactions (Supplementary Note 2). Simulations with different feedback strengths were carried out by varying the parameters quantifying binding strengths of active activator Ste12 (Ste12-P) to the respective promoters.

**Reporting summary**. Further information on research design is available in the Nature Research Reporting Summary linked to this article.

## Data availability

RNA sequencing data have been deposited in the Gene Expression Omnibus (GEO) database under accession code GSE151729. Further data supporting the findings of this study, including raw images, tables with single-cell data and flow cytometric data, are available upon request from the corresponding authors.

## Code availability

The code for simulations is available on request from the corresponding authors. Custom scripts for CellProfiler, MATLAB and R used for the analysis of data are likewise available upon request.

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

## Acknowledgements

We thank Pieter Rein ten Wolde, Giulia Malaguti and Sean Murray for insightful comments on the manuscript. This work was supported by grant 294761-MicRobE from the European Research Council.

## Author contributions

A.A., B.G. and V.S. conceived experiments; A.A. performed experiments; B.G. and A.A. analysed the data; B.G. performed computational modelling; T.G. performed mass spectrometric experiments; B.G., A.A. and V.S. wrote the paper.

## Competing interests

The authors declare no competing interests.
