## [Peer Review File · Nature Communications]

Reviewers' comments:

Reviewer #1 (Remarks to the Author):

This paper examines the relationship between noise and energy expenditure in the well-characterized, G protein-mediated, pheromone-response pathway of budding yeast. The data confirm that negative transcriptional feedback suppresses pathway noise and attempts to show a connection between noise, ATP consumption and growth fitness. The authors posit a tradeoff of signal optimization vs. the energetic cost of information processing. The concept is innovative and original. The data on noise, obtained from FACS analysis of a GFP transcription reporter, use established methods, are convincing but are largely confirmatory. The data on energy consumption are indirect and unconvincing and, consequently, the connection between the two remains unestablished. If a revised manuscript is to be considered, I would insist on some additional controls and on seeing some direct experimental measures of kinase phosphorylation, cellular ATP (abundance and flux) and alternative methods of altering cellular ATP, something other than mutation of a single redundant kinase in the pathway being measured.

Fig. 1 shows that activation of the pheromone response pathway leads to induction of pathway components. What is the logic of the colors used for panel A? I suggest red for negative regulators and green for positive regulators. The two red lines in panel B are difficult to distinguish. If the data are an average of the two measurements, that should be stated and preferably both data sets shown. It would be helpful to state that the cells lack bar1 and to include some validation that the bar1 + sst2 double mutant is healthy and free of suppressor mutations.

Fig. 2 shows that sst2 and phosphatase mutants are noisy. The sst2 mutant provides a nice control for noise. Why not test dig1 and fus3 deletions, which are also reported to suppress noise under basal and highly induced conditions, respectively? Analysis of dig1 would provide important information missing from the McCullagh Nature Communications paper. That paper states correctly that Dig1 suppresses the mating response and is transcriptionally induced, but it does NOT show that pheromone-dependent negative feedback by Dig1 leads to noise suppression, as claimed. Inexplicably they measure noise in unstimulated conditions, not in the presence of pheromone, so there is no evidence that DIG1 expression or negative feedback leads to noise suppression in that paper.

Related to this figure, the terms "Fisher information" and "Mutual information" will not be familiar to most biologists and the explanation provided is not helpful: "Precision is bounded by the Fisher information which measures uncertainty in estimating the input from the output distribution. Furthermore, the information transmission capacity of a signalling system can be quantified by calculating mutual information between the input and the output." If this concept is important to understanding the model, it should be clear to intelligent readers. Mathematical terms should be defined when used.

Fig. 3 shows that better induction of the negative regulator Msg5 leads to better suppression of signal and noise. This figure parallels Fig. 2 but with another promoter for MSG5. I recommend using the same layout in both figures, for easier comparison. The authors conclude from these experiments "that while the precision of input estimation could be significantly enhanced by artificially increasing the sensitivity of MSG5 gene induction, the naturally observed regulation appears optimal when considering energy investment into operation of the signalling pathway." On the other hand the differences could also be due to unequal expression from the two promoters. How is Msg5 *protein* expression affected by pheromone? or use of an alternative pheromone-inducible FUS1 promoter? or use of the doxycycline inducible promoter?

Fig. 5 Shows that two different inactivating mutations in the MAPK Fus3 have opposing effects on cell growth. That is, continuous phosphorylation and dephosphorylation of enzymatically inactive Fus3 lowers competitive growth fitness in the presence of pheromone. The K42R mutant is said to

be phosphorylated while KTY mutant is not. Is it really phosphorylated? Is it really dephosphorylated? Are the two proteins similarly induced? What is the role of the redundant MAPK Kss1, which in the absence of Fus3 activity will also transmit the pheromone response (leading to induction of Fus3, Sst2, Msg5, etc.). The mutant form of Fus3 lacks catalytic activity but it does not lack biological activity; in fact it inhibits pheromone signaling by unknown mechanisms (Nagiec 2015, PMID: 26179917). A good alternative here would be to repeat the experiment with mutants of Kss1, which in contrast to Fus3 is not induced and does not influence cell cycle progression (which could be confused with "growth fitness") and to swap the promoters for FUS3 and KSS1. An essential control here is to alter [ATP] by some method other than a point mutation in a single redundant kinase and to actually measure ATP flux in the cell. Otherwise the data do not support the model.

Reviewer #2 (Remarks to the Author):

The paper by Anders et al., addresses an interesting question of whether signal transduction might evolve to optimize information transfer subject to minimization of energetic cost. The paper uses the pheromone signaling response in baker's yeast, a frequently studied signaling system, to address this question. The authors argue that an increased expression of one of the phosphatases vs. the WT levels might increase information transfer, but at a higher cost to the cells. This conclusion is not tested directly, but the authors do experiments to suggest that the phosphorylation/dephosphorylation cycles carry additional fitness cost for the cells. Overall, although the question addressed is interesting, there many remaining question, and gaps in the authors' logic to warrant publication at this stage. Ultimately, the conclusions are only indirectly supported and the results, as they stand, not very surprising or revealing.

Specific concerns:

1. Many aspects of the analysis are poorly described. For instance, it is not clear how the promoter output is estimated. At what time is this analysis performed? Given that the authors analyze the FUS1 promoter using GFP, it is not clear why they did not estimate the output over the presumably multi-hour response. This would allow the analysis of how the information transfer might change over time, presumably as the protein expression of many of the pathway components also changes.
2. The calculation of the information metrics was not justified, or for that matter, discussed. Why did the authors choose to calculate the transfer for windows of three inputs. How are those windows chosen, given the nonlinear nature of the input-output responses. Why didn't the authors calculate the information capacity of the pathway (which would use all the input and output ranges), e.g., following the foundational study by Cheong et al, that the authors cite. This would allow them to contrast their results with those by others. The benefits of using the Fisher information vs. the usually employed mutual information are not discussed, and the discrepancy of the results (e.g., apparent maximization at different output values) is not touched upon.
3. Generally, as e.g., shown in Cheong et al., and other studies, the negative feedback can have two different effects in the mutual information capacity: 1) to reduce noise (shown in this study) and 2) reduce the dynamic range of the response (through an adaptive decrease of response amplitude). The first effect tends to increase the information capacity, whereas the second tends to decrease it. In this study, apparently, the negative feedback increases rather than decreases the dynamic range by inhibiting the baseline activity, but on affecting the maximal response. Why is the maximal response not affected? Again, here the analysis at different time points might be revealing.
4. The model presented by the authors appears to be poorly parameterized, particularly its detailed variant. How did the authors estimate the parameters? What are those parameters? Given that a lot of conclusions are based on modeling rather than experimentation, it is an important question. They are not the first to model this pathway or, for that matter, MAPK pathways. How

does their model compare to others? Would their conclusion translate to other systems?

5. The conclusions, as they stand, are not very surprising. As indicated above, it has been shown that changing the feedback strength can either decrease or increase information transfer, by perturbing the noise and dynamic range of the output. The authors do an interesting experiment towards the end of the study, in which they attempt to determine the fitness cost of Fus3 phosphorylation/dephosphorylation cycles. However, this experiment is only very indirectly related to the beginning of the study, and of course, the result is again not very surprising. This result does not specifically relate to the effects of Msg5 vs. many other phosphatases or the kinase, and does not provide a quantitative connection to the model (which would admittedly be very hard). Can the authors suggest a more direct test, e.g., evaluating the ATP consumption? Can they more directly study the effects of differential MSG5 expression (with appropriate controls)?

6. Overall, the study appears to be very disjoint. The first part focuses on the effects of Msg5, the second on development of a model, with rather general assumptions of how the energy is used in the pathway (or, rather, more generally in an abstract version of a signaling network), and the third on the interesting but somewhat unrelated experiment mentioned above. So, at the end the main thesis of the work is not really supported. The conclusion that signaling cost energy is not surprising. The statement that information per energy cost, rather than information itself can be evolutionarily optimized, although interesting, remains largely hypothetical.

Given the multiple concerns above, I cannot support the publication of this study in the present form.

Reviewer #1 (Remarks to the Author):

This paper examines the relationship between noise and energy expenditure in the well-characterized, G protein-mediated, pheromone-response pathway of budding yeast. The data confirm that negative transcriptional feedback suppresses pathway noise and attempts to show a connection between noise, ATP consumption and growth fitness. The authors posit a tradeoff of signal optimization vs. the energetic cost of information processing. The concept is innovative and original. The data on noise, obtained from FACS analysis of a GFP transcription reporter, use established methods, are convincing but are largely confirmatory. The data on energy consumption are indirect and unconvincing and, consequently, the connection between the two remains unestablished. If a revised manuscript is to be considered, I would insist on some additional controls and on seeing some direct experimental measures of kinase phosphorylation, cellular ATP (abundance and flux) and alternative methods of altering cellular ATP, something other than mutation of a single redundant kinase in the pathway being measured.

We thank the Reviewer #1 for finding the central concept of our manuscript innovative and original. We added several additional controls, including detection of kinase phosphorylation, which further support our conclusions that observed growth differences indeed reflect energy consumption by the pathway. Notably, however, direct measurements of the energy consumption by this single pathway at the ATP level would not be feasible, because it represents only a small fraction of the total cellular energy budget. This was precisely the reason why we used growth competition experiments to estimate the cost. See our detailed comments below.

Fig. 1 shows that activation of the pheromone response pathway leads to induction of pathway components. what is the logic of the colors used for panel A? I suggest red for negative regulators and green for positive regulators.

Colours in Fig. 1 are according to EC50 of induction, i.e., pheromone sensitivity of the induced feedback regulators. Different induction sensitivities of different feedback regulators (esp. the negative regulators operating at different pathway levels) is the main information that this figure should relay. This choice of colours was already illustrated in the figure itself, but now we also more clearly explain it in the legend. The logic of operation for different regulators (i.e., positive or negative) is depicted by arrows and lines, which is now again better explained in the legend.

The two red lines in panel B are difficult to distinguish. If the data are an average of the two measurements, that should be stated and preferably both data sets shown.

We now made the lines better distinguishable by using a dashed line for one of the fits. The data are not an average but, as the reviewer suggests, sets from two independent measurements. We now make this clearer by depicting results from different experiments with slightly different symbol styles (symbols with and without frames).

It would be helpful to state that the cells lack *bar1* and to include some validation that the *bar1* + *sst2* double mutant is healthy and free of suppressor mutations.

All strains in our study were deleted for *BAR1*. For better clarity, we now include this information in the main text and in the legend of Fig. 1 (formerly it was only stated in the MM section).

Fig. 1 itself does not contain data of a *bar1* + *sst2* double mutant, but Fig. 2 indeed does. As this reviewer correctly says (see below), it serves as a control for “noisy signalling”. The role of *SST2* in noise suppression is well established and was thus not the primary scope of this study. Our mutant shows the well-established and described phenotypes connected with deletion of *SST2*, i.e., strongly increased basal pathway activity, sensitivity and signalling noise. Indeed, *sst2* mutants are not “healthy” and grow slower than wild type, likely as a consequence of elevated basal pathway activity and consequent “faulty” strong expression of pheromone regulated genes (and possibly partial cell-cycle arrest). This does not affect any of our conclusions, but we now mention this growth effect of *SST2* deletion in supplementary table S1 describing the strains used in our study.

Fig. 2 shows that *sst2* and phosphatase mutants are noisy. The *sst2* mutant provides a nice control for noise. Why not test *dig1* and *fus3* deletions, which are also reported to suppress noise under basal and highly induced conditions, respectively?

All strains in our study were derived from strain SEY6210a that is devoid of active Kss1 (this is now stated specifically in the main text, not only in MM section), which was done to exclusively analyse the Fus3-branch of the signalling cascade and thus to simplify the analysis. Consequently, deletion of *FUS3* results in complete disruption of signalling downstream of Ste7. As for *DIG1* deletion, see below.

Analysis of *dig1* would provide important information missing from the McCullagh Nature Communications paper. That paper states correctly that Dig1 suppresses the mating response and is transcriptionally induced, but it does NOT show that pheromone-dependent negative feedback by Dig1 leads to noise suppression, as claimed. Inexplicably they measure noise in unstimulated conditions, not in the presence of pheromone, so there no evidence that *DIG1* expression or negative feedback leads to noise suppression in that paper.

To the best of our knowledge, *DIG1* has not been described to be transcriptionally induced by pheromone nor does it display transcriptional induction in our RNA-sequencing experiments. In the modified version of our manuscript we thus no longer refer to Dig1 as a potential negative feedback regulator (as was done based on McCullagh paper) anymore, and specifically mention this point in the Discussion. We thank the reviewer for pointing out the inconsistencies regarding this regulator.

Related to this figure, the terms “Fisher information and “Mutual information” will not be familiar to most biologists and the explanation provided is not helpful: “Precision is bounded by the Fisher information which measures uncertainty in estimating the input from the output distribution. Furthermore, the information transmission capacity of a signalling system can be quantified by calculating mutual information between the input and the output.” If this concept is important to understanding the model, it should be clear to intelligent readers. Mathematical terms should be defined when used.

We apologize for being too brief here. In addition to the description of terminology in the SI, we now expanded the Introduction (and Discussion) to better explain these concepts in the main text and also defined the mathematical terms where appropriate.

Fig. 3 shows that better induction of the negative regulator *Msg5* leads to better suppression of signal and noise. This figure parallels Fig. 2 but with another promoter for *MSG5*. I recommend using the same layout in both figures, for easier comparison.

Fig.3 only partially parallels Fig. 2, because it additionally contains a panel for simulations and lacks the panel with mutual information (2D). So making both figures to have the same layout would be difficult.

The authors conclude from these experiments “that while the precision of input estimation could be significantly enhanced by artificially increasing the sensitivity of *MSG5* gene induction, the naturally observed regulation appears optimal when considering energy investment into operation of the signalling pathway.” On the other hand the differences could also be due to unequal expression from the two promoters.

How is *Msg5* *protein* expression affected by pheromone?

or use of an alternative pheromone-inducible *FUS1* promoter?

We did test several promoters all of which confer more sensitive induction than the native *MSG5* promoter (Fig. S7, formerly Fig. S6) and with any of those promoters controlling *MSG5* expression the precision of input estimation was enhanced. We also tested expression of *MSG5* by doxycycline inducible promoter (Fig. S5, formerly Fig. S4), confirming that the feedback regulation is important to achieve low noise levels and thus high information as seen for the wild type. In this case, even with a dose-response curve (i.e., output range) similar to that of wild type, Fisher information remained much lower due to missing feedback regulation and consequently increased noise.

However, the reviewer is correct in pointing out that (absolute) expression levels of the feedback regulators may influence the precision of input estimation independent of their EC_{50} of induction. Experimentally, we were not able to separate induction sensitivity from absolute expression levels, since all promoters tested to drive pheromone-dependent expression of *MSG5* conferred both higher sensitivity and higher expression levels, as evidenced by lower saturated P_{FUS1} output (Fig. S7). Likewise, in our simulations, not only sensitivity of gene induction but also expression levels were changed by changing KD values of Ste12 binding to the gene's promoters. Nevertheless, we now performed additional analysis to estimate contributions of output range (influenced by expression levels of negative regulators) and noise (influenced by sensitivity of feedback induction) for our promoter exchange experiments (Fig. S8). These analyses confirmed that both output range and noise reduction at low pathway activity contribute to increased Fisher information in strains with sensitized *MSG5* induction. We now make this clear in the modified version of the manuscript.

Fig. 5 Shows that two different inactivating mutations in the MAPK *Fus3* have opposing effects on cell growth. That is, continuous phosphorylation and dephosphorylation of enzymatically inactive *Fus3* lowers competitive growth fitness in the presence of pheromone. The K42R mutant is said to

be phosphorylated while KTY mutant is not. Is it really phosphorylated? Is it really dephosphorylated?

We would like to emphasize that both Fus3 variants carry the same inactivating mutation (K42R), with KTY carrying additional mutations in the phosphorylation sites. Fus3-K42R has been previously shown to be phosphorylated *in vivo* (Gartner et al. Genes Dev. 1992; Nagiec 2015 mentioned by reviewer below), but we have nevertheless performed additional experiments to directly confirm its pheromone-dependent phosphorylation in strains used for the competition experiments using mass spectrometry. Notably, this phosphorylation was stronger in a *msg5* deletion strain, thus providing support for *in-vivo* phosphorylation as well as Msg5 dependent de-phosphorylation of Fus3-K42R. These new results are now shown in supplemental Fig. S12. We further modified the main text to refer to this control experiment.

Are the two proteins similarly induced?

Our Western blot analysis (Fig S11B, formerly Fig. S9B) suggests that there is indeed no bias in expression between K42R and KTY. Even more importantly, in our growth competition experiments, we measure ratios between cells expressing either protein variant in the presence of pheromone and normalize those to ratios obtained in the absence of pheromone. Thus, growth differences must be specifically caused by pheromone and cannot be just the result of different protein expression levels (since those are likely to influence growth in absence AND presence of pheromone likewise). Further control experiments which we have now added, support this point: After deletion of *Ste7*, K42R- and KTY-expressing strains no longer display pheromone dependent growth difference (Fig 5C), confirming that growth deficiency of K42R is dependent on its phosphorylation and could not be explained by different expression levels compared to KTY.

What is the role of the redundant MAPK Kss1, which in the absence of Fus3 activity will also transmit the pheromone response (leading to induction of Fus3, Sst2, Msg5, etc.).

As mentioned above, our strain lacks active Kss1, which is now explained in the Results and not only in the Methods. Thus, no pheromone signal is transmitted further downstream in the absence of (catalytically active) Fus3. Consistently, we have measured pheromone response of our strains with different Fus3 variants (based on P_{FUS1} -mCherry reporter) and were not able to see any induction of the reporter gene (see Fig S11C, formerly Fig 9C). We have now added plots below the main plots which display zoomed-in data for reporter gene expression in those strains to emphasize this finding.

The mutant form of Fus3 lacks catalytic activity but it does not lack biological activity; in fact it inhibits pheromone signaling by unknown mechanisms (Nagiec 2015, PMID: 26179917).

It is our understanding that mutant forms of Fus3 (in the reference mentioned those are actually single/double phosphorylation mutants, but not catalytically inactive mutants) can affect signalling activity by interfering with otherwise active signalling, i.e., when expressed along another copy of active/ wild type Fus3 (or Kss1). In contrast, our strains used for growth-competition experiments are devoid of any signal transmission downstream of *Ste7* and reporter gene expression (see above).

A good alternative here would be to repeat the experiment with mutants of *Kss1*, which in contrast to *Fus3* is not induced and does not influence cell cycle progression (which could be confused with “growth fitness”) and to swap the promoters for *FUS3* and *KSS1*.

Expression of *Fus3* mutants in competition experiments was controlled by doxycycline and thus was not subjected to transcriptional auto-regulation. This renders *Fus3* expression constitutive and thus equivalent to a promoter-swap suggested by the reviewer.

Regarding potential effects of cell-cycle arrest: As noted above, we see no indication of any signalling activity of *Fus3*-K42R (or -KTY). Nevertheless, we performed an additional control (shown in Fig. 5C) by repeating the competition experiment but in a strain deleted for *MSG5*. We observed significantly decreased growth difference between *Fus3*-K42R and -KTY in this background, consistent with our interpretation that growth differences are due to the cost of phosphorylation/dephosphorylation cycle (which should be slower in this mutant background) and not due to cell-cycle arrest (which should be stronger in this background, since it has higher level of *Fus3* phosphorylation).

An essential control here is to alter [ATP] by some method other than a point mutation in a single redundant kinase and to actually measure ATP flux in the cell. Otherwise the data do not support the model.

[see also related part of the reviewers introducing paragraph: “If a revised manuscript is to be considered, I would insist on some direct experimental measures of ... cellular ATP (abundance and flux) and alternative methods of altering cellular ATP, something other than mutation of a single redundant kinase in the pathway being measured.”]

We would like to point out that we are not aiming at the general investigation of the relation between alterations of the cellular ATP concentration and growth fitness, but rather to quantify the fitness effect of the phosphorylation/de-phosphorylation cycle for a specific kinase *Fus3*. While it is clear that this cycle will consume energy, the expected energetic cost of this cycle is marginal compared to total cellular energy/ ATP turnover. Thus, measuring this cost by quantifying the total levels (or fluxes) of ATP would not be feasible, and this is exactly the reason why long-term competition experiments were required to detect this cost. Experimental demonstration that energy consumed in this process, while being seemingly marginal compared to overall cellular consumption, could nevertheless be significant enough to affect cellular fitness and thus being subject to evolutionary optimization in order to balance cost (energy) and benefit (precision) is one of the key points of our manuscript. To our knowledge, this is the first time that the effect of a noise-suppressing reaction cycle on cellular fitness has been demonstrated.

Reviewer #2 (Remarks to the Author):

The paper by Anders et al., addresses an interesting question of whether signal transduction might evolve to optimize information transfer subject to minimization of energetic cost. The paper uses the pheromone signaling response in baker’s yeast, a frequently studied signaling system, to address this question. The authors argue that an increased expression of one of the phosphatases vs. the WT levels might increase information transfer, but at a higher cost to the cells. This conclusion is not

tested directly, but the authors do experiments to suggest that the phosphorylation/dephosphorylation cycles carry additional fitness cost for the cells. Overall, although the question addressed is interesting, there many remaining questions, and gaps in the authors' logic to warrant publication at this stage. Ultimately, the conclusions are only indirectly supported and the results, as they stand, not very surprising or revealing.

We thank the Reviewer #2 for finding the topic of our manuscript interesting and for helpful suggestions on improving the manuscript. We now performed additional experiments and analyses to support and sharpen our conclusions, and improved description of the experiments and modelling.

Specific concerns:

Many aspects of the analysis are poorly described. For instance, it is not clear how the promoter output is estimated. At what time is this analysis performed? Given that the authors analyze the FUS1 promoter using GFP, it is not clear why they did not estimate the output over the presumably multi-hour response. This would allow the analysis of how the information transfer might change over time, presumably as the protein expression of many of the pathway components also changes.

The promoter output was estimated by measuring fluorescence expressed from a P_{FUS1} -GFP reporter gene (except Fig. 1). We slightly modified the main text to make it even clearer that this fluorescence was the output. Data shown in Fig. 2 were acquired between 140 and 210 min, as mentioned in the corresponding Figure legend, but similar results were obtained at other time intervals that are now shown in Fig. S4. While the exact values of Fisher and mutual information are time-dependent, the overall trend remains the same, particularly at later time points.

2. The calculation of the information metrics was not justified, or for that matter, discussed. Why did the authors choose to calculate the transfer for windows of three inputs. How are those windows chosen, given the nonlinear nature of the input-output responses. Why didn't the authors calculate the information capacity of the pathway (which would use all the input and output ranges), e.g., following the foundational study by Cheong et al, that the authors cite. This would allow them to contrast their results with those by others. The benefits of using the Fisher information vs. the usually employed mutual information are not discussed, and the discrepancy of the results (e.g., apparent maximization at different output values) is not touched upon.

We sincerely apologize for insufficient description of the information metrics and their significance in the main text. This is now improved and differences between Fisher and mutual information, as well as our rationale for primarily using Fisher information and calculating mutual information in the intervals of inputs (essentially, because the trade-off between information transmission and corresponding energetic cost is defined for a particular input) is better explained in Introduction, Results and Discussion. Moreover, while redoing the calculation of Fisher information, we spotted a mistake in alignment of the values in the plot (for which we apologize), and corrected values show similar dependence on the output for both types of information. Importantly, our conclusions remain the same regardless whether Fisher or mutual information is used, or when summing up the information over the entire range of inputs.

3. Generally, as e.g., shown in Cheong et al., and other studies, the negative feedback can have two different effects in the mutual information capacity: 1) to reduce noise (shown in this study) and 2) reduce the dynamic range of the response (through an adaptive decrease of response amplitude). The first effect tends to increase the information capacity, whereas the second tends to decrease it. In this study, apparently, the negative feedback increases rather than decreases the dynamic range by inhibiting the baseline activity, but on affecting the maximal response. Why is the maximal response not affected? Again, here the analysis at different time points might be revealing.

The reviewer is correct in pointing out that negative feedback can have a negative effect on the output range (by decreasing maximal output) and thus on mutual information. However, as shown before (e.g., at earlier time point in the cited study) it can also increase the output range by suppressing basal pathway activity, and thus increase information in addition to suppression of noise. We now modified the Introduction and Discussion to mention both possibilities. Which of these regimes is realized in a specific situation depends on an interplay between basal activity and feedback strength. In the system analysed by us, the effect on basal activity is apparently more crucial and the output range is increased by negative feedback (though this effect seems to become weaker at later time points, see Fig. S4). Regarding the question about the unaffected maximal response: It seems likely that the pathway output / activity of the transcriptional reporter is already saturated at sub-saturation Fus3-PP levels. Thus, further increase of Fus3-PP upon deletion of negative feedbacks would not increase pathway output.

4. The model presented by the authors appears to be poorly parameterized, particularly its detailed variant. How did the authors estimate the parameters? What are those parameters? Given that a lot of conclusions are based on modeling rather than experimentation, it is an important question. They are not the first to model this pathway or, for that matter, MAPK pathways. How does their model compare to others? Would their conclusion translate to other systems?

We now better describe how the parameters of the model were selected and also list them in the SI text when describing the model. Essentially, the parameters have been chosen based on previous publications and adjusted where necessary to reproduce our experimental data. Importantly, our model is substantially simpler compared to previously published models of the mating pathway, so many parameter values are not directly comparable. Nevertheless, as supported by our analytical calculations in the SI text, we argue that our key conclusion that the symmetry of information with respect to the dissipation rates is broken when we quantify information per unit energy is general and does not qualitatively depend on specific details or parameters of the model.

5. The conclusions, as they stand, are not very surprising. As indicated above, it has been shown that changing the feedback strength can either decrease or increase information transfer, by perturbing the noise and dynamic range of the output.

We agree that description of a negative feedback being able to influence noise and output range is on itself not surprising (though it has not been shown explicitly for Msg5 before). However, our ability to reduce the noise below wild type levels (at least in certain output range) by simply changing the feedback strength was rather surprising to us and to our knowledge has not been

demonstrated before. Most importantly, the conclusion that the induction order of feedback regulators appears to be optimized for a trade-off between information and energy rather than information alone is not trivial at all and it has general significance.

The authors do an interesting experiment towards the end of the study, in which they attempt to determine the fitness cost of Fus3 phosphorylation/dephosphorylation cycles. However, this experiment is only very indirectly related to the beginning of the study, and of course, the result is again not very surprising.

First, we would like to thank the reviewer for appreciating the experiment, which indeed for the first time demonstrates *in vivo* the fitness cost of a reaction cycle involved in signalling. It is complementing the first part of the study, so that we overall characterize both improved information transmission and increased fitness cost associated with negative (feedback) regulators. We have now modified the text to better discuss this connection, and we performed additional control experiments to make the quantification of the fitness cost even more compelling.

This result does not specifically relate to the effects of Msg5 vs. many other phosphatases or the kinase, and does not provide a quantitative connection to the model (which would admittedly be very hard).

Our new experiments (added as Fig. 5C) confirm the effect of Msg5 on the observed fitness cost. Direct comparison of the experimentally observed fitness cost to the ATP consumption in the model would be indeed hard, since the predicted energy consumption in this particular reaction is comparatively small and cannot be directly quantified (see our reply to comments of Reviewer #1).

Can the authors suggest a more direct test, e.g., evaluating the ATP consumption? Can they more directly study the effects of differential MSG5 expression (with appropriate controls)?

As mentioned above and in our reply to Reviewer #1, the expected ATP consumption in this one reaction is very small compared to the total cellular energy budget, and it is already a major advance to show that it has a measurable fitness cost. Regarding the second point, we now performed additional experiments (shown in Fig. 5C) to confirm that the fitness cost depends on Fus3 dephosphorylation by Msg5, since the cost diminishes by roughly 50% upon deletion of *MSG5*. Furthermore, we now demonstrate that prevention of Fus3-K42R phosphorylation by deletion of upstream kinase Ste7 completely abolishes the pheromone-specific growth defect of Fus3-K42R-expressing strains.

6. Overall, the study appears to be very disjoint. The first part focuses on the effects of Msg5, the second on development of a model, with rather general assumptions of how the energy is used in the pathway (or, rather, more generally in an abstract version of a signaling network), and the third on the interesting but somewhat unrelated experiment mentioned above. So, at the end the main thesis of the work is not really supported. The conclusion that signaling cost energy is not surprising. The statement that information per energy cost, rather than information itself can be evolutionarily optimized, although interesting, remains largely hypothetical.

We now extensively modified the main text in order to make it more conclusive and to improve the connections between its different parts clearer.

REVIEWERS' COMMENTS:

Reviewer #1 (Remarks to the Author):

The revised manuscript is substantially improved with some important clarifications to the text and figures, the addition of new requested data, some new and interesting comparisons between GTP and ATP energy expenditure, and a more narrow set of conclusions.

The most substantive issue for me was the purported connection between ATP consumption and fitness. As I noted in my previous review, "if a revised manuscript is to be considered, I would insist on some additional controls and on seeing some direct experimental measures of ... cellular ATP (abundance and flux) and alternative methods of altering cellular ATP, something other than mutation of a single redundant kinase in the pathway being measured." Reviewer 2 expressed a similar concern.

The authors appear to have softened their claims but the stated reason for not doing the experiment does not seem logical to me. According to the response, "direct measurements of the energy consumption by this single pathway at the ATP level would not be feasible, because it represents only a small fraction of the total cellular energy budget. This was precisely the reason why we used growth competition experiments to estimate the cost." If I understand the argument, the changes in ATP are supposedly too small to be measured (or even attempted) but sufficiently large to limit cell growth? I don't buy it, and in the absence of experimental evidence, I would recommend rewriting the conclusions, shifting the emphasis from global energy expenditure to nonproductive phosphorylation and dephosphorylation of the MAPK Fus3.

The data do not support that broad central claim, that they have demonstrated "for the first time experimentally that dissipation of energy during signalling indeed can bear measurable fitness cost and thus is likely to be under selection for trade off optimization between information and energy consumption."

I see two solutions. The preferred is to measure ATP. In such case it would be good to include in all of the experiments a positive control, for example introducing a non-productive ATPase, perhaps with non-yeast enzymes, and measuring the pheromone response.

The easier (and to me acceptable) alternative is to avoid ascribing the noise to a mechanism that is not experimentally tested and claiming otherwise. I would be satisfied if mechanistic conclusions, ascribing the effects on growth to energy expenditure, were replaced with something closer to what is actually tested. For example, claiming instead that they have shown "for the first time experimentally that nonproductive cycling of MAPK phosphorylation and dephosphorylation can bear measurable fitness cost and thus is likely to be under selection for trade off optimization between information and local energy consumption." That is almost as interesting and far more accurate in my view.

Manuscript NCOMMS-19-08753A

Point-by-point response to reviewer's comments

Reviewer #1 (Remarks to the Author):

The revised manuscript is substantially improved with some important clarifications to the text and figures, the addition of new requested data, some new and interesting comparisons between GTP and ATP energy expenditure, and a more narrow set of conclusions.

The most substantive issue for me was the purported connection between ATP consumption and fitness. As I noted in my previous review, "if a revised manuscript is to be considered, I would insist on some additional controls and on seeing some direct experimental measures of ... cellular ATP (abundance and flux) and alternative methods of altering cellular ATP, something other than mutation of a single redundant kinase in the pathway being measured." Reviewer 2 expressed a similar concern.

The authors appear to have softened their claims but the stated reason for not doing the experiment does not seem logical to me. According to the response, "direct measurements of the energy consumption by this single pathway at the ATP level would not be feasible, because it represents only a small fraction of the total cellular energy budget. This was precisely the reason why we used growth competition experiments to estimate the cost." If I understand the argument, the changes in ATP are supposedly too small to be measured (or even attempted) but sufficiently large to limit cell growth? I don't buy it, and in the absence of experimental evidence, I would recommend rewriting the conclusions, shifting the emphasis from global energy expenditure to nonproductive phosphorylation and dephosphorylation of the MAPK Fus3.

The data do not support that broad central claim, that they have demonstrated "for the first time experimentally that dissipation of energy during signalling indeed can bear measurable fitness cost and thus is likely to be under selection for trade off optimization between information and energy consumption."

I see two solutions. The preferred is to measure ATP. In such case it would be good to include in all of the experiments a positive control, for example introducing a non-productive ATPase, perhaps with non-yeast enzymes, and measuring the pheromone response.

The easier (and to me acceptable) alternative is to avoid ascribing the noise to a mechanism that is not experimentally tested and claiming otherwise. I would be satisfied if mechanistic conclusions, ascribing the effects on growth to energy

expenditure, were replaced with something closer to what is actually tested. For example, claiming instead that they have shown “for the first time experimentally that nonproductive cycling of MAPK phosphorylation and dephosphorylation can bear measurable fitness cost and thus is likely to be under selection for trade off optimization between information and local energy consumption.” That is almost as interesting and far more accurate in my view.

We thank the Reviewer for acknowledging that the revised manuscript is substantially improved. As stated already in our previous response, we do not believe that direct measurement of the energetic cost of the phosphorylation/de-phosphorylation cycle of Fus3 (estimated to be around 0.2% of total cellular ATP flux) is experimentally feasible, but this estimated energetic cost does agree very well with the measured effect of this cycle on growth fitness. Following suggestion of this Reviewer, we have now modified Abstract and Discussion to be more specific and emphasize that what we experimentally measured is the fitness cost of futile cycling within this particular MAPK cascade:

Abstract: “We further demonstrate that futile cycling of MAPK phosphorylation and dephosphorylation has a measurable effect on growth fitness, with energy dissipation within the signalling cascade thus likely being subject to evolutionary selection.”

Discussion: “...we further demonstrate experimentally that futile cycling of Fus3 phosphorylation and dephosphorylation indeed can bear measurable fitness cost. Thus, dissipation of energy at this step likely resulted in evolutionary selection for trade-off optimization between information and energy consumption.”